# Host Responses Following Infection with Canadian-Origin Wildtype and Vaccine Revertant Infectious Laryngotracheitis Virus

**DOI:** 10.3390/vaccines10050782

**Published:** 2022-05-16

**Authors:** Esraa A. Elshafiee, Ishara M. Isham, Shahnas M. Najimudeen, Ana Perez-Contreras, Catalina Barboza-Solis, Madhu Ravi, Mohamed Faizal Abdul-Careem

**Affiliations:** 1Department of Ecosystem and Public Health, Faculty of Veterinary Medicine, University of Calgary, 3330 Hospital Drive NW, Calgary, AB T2N 4N1, Canada; esraa.elshafiee@ucalgary.ca (E.A.E.); fathimaishara.muhamm@ucalgary.ca (I.M.I.); fathimashahnas.moham@ucalgary.ca (S.M.N.); a.perez@morphocell.com (A.P.-C.); catalina.barboza-solis@mcgill.ca (C.B.-S.); 2Department of Zoonoses, Faculty of Veterinary Medicine, Cairo University, Giza 12211, Egypt; 3Animal Health and Assurance, Alberta Agriculture and Forestry, Edmonton, AB T6H 4P2, Canada; madhubabu.ravi@crl.com

**Keywords:** infectious laryngotracheitis virus (ILTV), poultry, pathogenesis, host response, vaccine revertant infectious laryngotracheitis virus, backyard poultry, Canada

## Abstract

Infectious laryngotracheitis (ILT) is caused by Gallid herpesvirus-1 (GaHV-1) or infectious laryngotracheitis virus (ILTV) and was first described in Canadian poultry flocks. In Canada, ILTV infection is endemic in backyard flocks, and commercial poultry encounters ILT outbreaks sporadically. A common practice to control ILT is the use of live attenuated vaccines. However, outbreaks still occur in poultry flocks globally due to ILTV vaccine strains reverting to virulence and emergence of new ILTV strains due to recombination in addition to circulating wildtype strains. Recent studies reported that most of the ILT outbreaks in Canada were induced by the chicken-embryo-origin (CEO) live attenuated vaccine revertant strains with the involvement of a small percentage of wildtype ILTV. It is not known if the host responses induced by these two ILTV strains are different. The objective of the study was to compare the host responses elicited by CEO revertant and wildtype ILTV strains in chickens. We infected 3-week-old specific pathogen-free chickens with the two types of ILTV isolates and subsequently evaluated the severity of clinical and pathological manifestations, in addition to host responses. We observed that both of the isolates show high pathogenicity by inducing several clinical and pathological manifestations. A significant recruitment of immune cells at both 3 and 7 days post-infection (dpi) was observed in the tracheal mucosa and the lung tissues of the infected chickens with wildtype and CEO vaccine revertant ILTV isolates when compared to uninfected controls. Overall, this study provides a better understanding of the mechanism of host responses against ILTV infection.

## 1. Introduction

Infectious laryngotracheitis (ILT) is caused by infectious laryngotracheitis virus (ILTV, Gallid herpesvirus 1), a member of the *Alphaherpesvirinae* sub-family and the genus *Iltovirus*, and it infects the upper respiratory tract of chickens [1]. In addition to chickens [2], peafowl and pheasants have also been reported to be infected with this virus [3,4].

ILTV infection causes significant economic losses for the poultry sector around the world due to high mortalities and production losses [5]. The severity of the clinical presentation of the disease is dependent on the infecting ILTV strain and is identified by weight loss, respiratory distress, nasal discharge, sinusitis, and conjunctivitis [6]. Gross pathological lesions include fibrinous and/or hemorrhagic exudates with necrosis and the ulceration of mucosal lining of the upper respiratory tract. Furthermore, the presence of hemorrhagic and fibrinous exudate could clog the airways and thereby cause gasping and death from asphyxia [7]. ILTV initially replicates within the epithelium of the laryngeal, tracheal, and conjunctival mucosa, resulting in histopathological changes such as the sloughing and necrosis of the mucosal epithelium with the formation of syncytial cells and eosinophilic intranuclear viral inclusions admixed with fibrinous and/or hemorrhagic exudate [1,8]. Furthermore, the thickening of the tracheal mucosa due to the intense infiltration of lymphocytes, heterophils, and macrophages could also be seen histologically [9,10,11]. The virus load peaks between 4 and 6 days post-infection (dpi) at the primary replication sites, and then the virus can be found at the predominant latency site, the trigeminal ganglia, resulting in the lifelong carrier stage [2,12,13]. Horizontal transmission through the respiratory tract, the oral cavity, or the ocular mucosa is the most common method of ILTV transmission [14,15]; however, mechanical transmission may also occur through vectors [5,16].

Strict biosecurity measures along with vaccination are the main strategies employed for ILT prevention and control [17]. Presently, vaccination is done with live attenuated vaccines and viral vector recombinant vaccines. The live attenuated vaccines are still acceptable choices for managing the disease as they provide reliable protection against viral infection by reducing the clinical signs and viral replication more efficiently than the recombinant vaccines [17,18,19,20,21]. Despite their efficiency, attenuated vaccines, especially chicken-embryo-origin (CEO) vaccines, have adverse properties of regaining virulence due to bird-to-bird passages, resulting in the development of virulent ILTV vaccinal strains in the field [22]. The recombination of different ILTV strains, including vaccine strains, gives rise to the emergence of highly virulent field strains [23,24,25]. Several studies based on ILTV outbreaks in various geographical regions report that a large number of ILTV isolated in the field were classified as vaccine revertant strains [26,27].

ILTV glycoproteins tend to be the most immunogenic, which is capable of inducing both B cell and T cell-mediated immune responses [28]. However, protection from ILT is mediated by the cell-mediated immune response rather than the antibody-mediated immune response [28,29,30,31]. The experiments using bursectomized chickens showed that ILTV antibodies are not protective [28,29], and it has been demonstrated in thymectomized chickens that cell-mediated immunity is critical against ILT [30]. Additionally, a correlation has been observed between increased pro-inflammatory and anti-inflammatory cytokine gene transcription and enhanced inflammatory cell recruitment, significant tissue damage, and decreased ILTV replication in the trachea [32]. 

In Canada, the genomic surveillance of the circulating ILTV isolates done by most recent studies, using partial- and whole-genome sequencing, reported >80% prevalence of CEO vaccine revertant ILTV isolates linked to ILT outbreaks [25,33]. The CEO vaccine revertant ILTV strains are reported as more pathogenic than the wildtype ILTV [13,34,35]; however, the available data about the pathogenesis of the Canadian isolates is still limited. We also do not know if CEO revertant and wildtype ILTV elicit differential host responses in chickens. Hence, the purpose of this study was to study the pathogenesis of, and host responses to, Canadian-origin CEO revertant and wildtype ILTV infection in chickens in a comparative manner.

## 2. Materials and Methods

### 2.1. Animals and Eggs

The approval for the study was obtained from the Health Science Animal Care Committee (HSACC) of the University of Calgary (Protocol number: AC19-0013). SPF eggs for the virus propagation and titration and chickens for the in vivo experiment were purchased from the Canadian Food Inspection Agency (CFIA), Ottawa, ON, Canada. 

### 2.2. Virus

The ILTV isolates originated from ILT outbreaks in backyard flocks of Alberta and were characterized based on whole-genome sequences as wildtype (AB-S63) and CEO vaccine revertant (AB-S45) [25]. As described previously, the chorioallantoic membrane (CAM) of 9–11-days-old SPF chicken embryos and monolayers of chicken embryo liver cells (CELICs) were used to propagate the AB-63 and AB-45 ILTV isolates, respectively [18,25]. The ILTV isolates were titrated in 96-well plates of CELICs, and the 50% tissue culture infective dose (TCID50) was determined as per the Reed and Muench method [36].

### 2.3. Experimental Design

Chickens were randomly divided into three groups each consisting of 11 birds at 3 weeks of age. The first group was infected with wildtype AB-S63 ILTV isolate, the second group was infected with CEO vaccine revertant AB-S45 ILTV isolate, and the third group was maintained as the uninfected control. Chickens were inoculated via both the intratracheal and intraocular routes with a dose of 10^3.5^ tissue culture infectious dose (TCID)_50_ in a total volume of 200 μL per chicken. The third group was mock-infected with 200 μL of sterile phosphate-buffered saline (PBS) via the same routes. The infected group was placed in high containment poultry isolators (Plas Labs Inc., Lansing, MI, USA) at the Prion Virology Facility (PVF), Foothills Campus, University of Calgary.

The chickens were monitored twice daily for clinical signs consistent with ILT, as has been described previously [18,34]. At 3 and 7 dpi, the bodyweights of the birds were recorded and oropharyngeal (OP) and cloacal (CL) swab samples were collected using sterile polyester swabs in universal transport medium (Puritan, Guilford, ME, USA) and stored at −80 °C until processed.

At 3 dpi, five chickens were euthanized from each group, while the remaining birds (6 per group) were observed until 7 dpi, when they were euthanized. Necropsies were performed to record the gross lesions and the sample collection. Tracheal samples for histopathological examination were collected in 10% buffered formalin (VWR International, West Chester, PA, USA). In addition, tissue samples were collected in RNA Save (Biological Industries, FroggaBio, Toronto, ON, Canada).

### 2.4. Techniques

#### 2.4.1. DNA Extraction

DNA extraction from swabs and tissue samples (trachea, and lungs) for ILTV genome load quantification was done using QIAmp DNeasy Kit (QIAGEN GmbH, Hilden, Mettmann, Germany) according to the manufacturer’s instructions. Total DNA was extracted from 200 μL of swab and approximately 50 mg of tissue samples. Then, the quantity and purity of the extracted DNA was evaluated using the Nanodrop ND-1000 spectrophotometer (Thermo Scientific, Wilmington, DE, USA).

#### 2.4.2. RNA Extraction and Reverse Transcription

RNA extraction from trachea and lungs was performed using the Trizol (Ambion, Invitrogen Canada Inc., Burlington, ON, Canada) reagent following manufacturer’s protocol. The concentration and purity of the extracted RNA was measured using the Nanodrop 1000 spectrophotometer (Thermo Scientific, Wilmington, DE, USA). Complementary (c)DNA synthesis from approximately 2000 ng of tissue samples was performed using RT random primers (high-capacity cDNA reverse transcriptase kit, Invitrogen Life Technologies, Carlsbad, CA, USA).

#### 2.4.3. ILTV Genome Load Quantification and Cytokine mRNA Expression by Quantitative (q)PCR Assays

The qPCR assay was done using the CFX96-c1000 Thermocycler (Bio-Rad laboratories, Mississauga, ON, Canada). The total reaction volume was maintained at 20 μL. Each reaction included genomic DNA or cDNA as a template, 10 μL of SYBR Green Master Mix (Invitrogen, Burlington, ON, Canada), 0.5 μL of 10 pmol/μL forward (F)- and reverse (R)-specific primers, and DNAse/RNAse-free water (Thermo Scientific, Wilmington, DE, USA). The primers for ILTV (proteinase kinase gene), interferon (IFN)-γ, enzyme inducible nitric oxide synthase (iNOS), and proinflammatory cytokine interleukin (IL)-1β encoding genes were described previously [18,37,38,39,40]. The thermocycler conditions were initial denaturation at 95 °C for 20 s, following 40 cycles of denaturation at 95 °C for 3 s, annealing at 60 °C for 30 s, and elongation at 95 °C for 10 s. The mRNA expressions of cytokines were quantified in relation to the β-actin housekeeping gene using the Pfaffl method [37,39].

#### 2.4.4. Histopathology and Image Analysis

Formalin-fixed tracheal tissues were processed (paraffin-embedded; sectioned; and hematoxylin and eosin-stained) by the Diagnostic Services Unit (DSU) of the University of Calgary, Faculty of Veterinary Medicine. The stained tracheal sections were examined, and tracheal lesion severity scoring was performed based on a previously reported scoring system developed by Guy and colleagues [9]. The measurement of mucosal thickness was performed using Aperio ImageScope (version 12.4.3.5008; Leica Biosystems, Vista, CA, USA) image analysis software. The method used for the measurement of mucosal thickness was adapted from the procedure described by Wilson and colleagues [41].

#### 2.4.5. Immunofluorescent Assay

Immunofluorescence staining of immune cells such as CD4+ T cells, CD8+ T cells, and macrophages was done as previously described [37,42].

### 2.5. Data and Statistical Analyses

The quantification of ILTV genome loads was done using the standard curves of the proteinase kinase gene plasmid and of the housekeeping gene, β-actin. An analysis of body weight data was done using a two-way analysis of variance (ANOVA) followed by Tukey’s multiple comparison test. The Kruskal–Wallis test followed by Dunn’s multiple comparison tests were used for the clinical score and the viral genome load analyses. The Mann–Whitney U test was used to evaluate the differences between CD4+ T cells, CD8+ T cells, macrophages, and cytokines mRNA expression at 3 and 7 dpi. All statistical analyses in this study were carried out using GraphPad Prism 9.0.0 (GraphPad Software, San Diego, CA, USA).

## 3. Results

### 3.1. Clinical Signs

The peak of clinical manifestations within the two infected groups was mirrored by the clinical scores from 3 to 5 dpi clearly (Figure 1a), while chickens in control group showed no clinical signs. At 3 and 4 dpi, the clinical score of AB-S63 ILTV was significantly higher than AB-S45 ILTV (*p* = 0.0005, *p* = 0.0033) and controls (*p* < 0.0001). Moreover, at 5 dpi, a significant difference in clinical signs was observed between AB-S63 ILTV (*p* < 0.0001), AB-S45 ILTV (*p* = 0.0039), and the controls. There was no difference in the bodyweights of the chickens infected with the wildtype ILTV (AB-S63) and CEO revertant ILTV (AB-S45) and the controls, and this remained constant throughout the experiment (Figure 1b, *p* > 0.05).

### 3.2. ILTV Genome Loads

The Figure 2 presents the ILTV genome loads in OP and CL swabs of experimental chickens. The ILTV genome loads in OP swabs of the CEO revertant ILTV (AB-S45)-infected group was higher than that of the wildtype (AB-S63) ILTV-infected group (*p* < 0.05) at 3 and 7 dpi (Figure 2a). There were no significant differences in the ILTV genome loads in the CL swabs between the two infected groups at 3 dpi (*p* > 0.05); however, at 7 dpi, the ILTV genome loads in the wildtype (AB-S63) ILTV-infected group were higher than those of the CEO revertant ILTV (AB-S45)-infected group (*p* < 0.001) (Figure 2b). No viral load was identified in the OP and CL swabs of the control group. At 3 dpi, the ILTV genome load in the trachea of the chickens infected with CEO revertant ILTV (AB-S45) was higher than that in the chickens infected with wildtype (AB-S63) ILTV (*p* < 0.001), whereas at 7 dpi, the differences in the ILTV genome loads in the trachea of the two infected groups were not significant (*p* > 0.05), although the ILTV genome load in the trachea of chickens infected with CEO revertant ILTV (AB-S45) was higher than the controls (*p* < 0.05) (Figure 2c). At 3 and 7 dpi, no significant difference in ILTV genome loads in the lungs between the two experimentally infected groups (*p* < 0.05) (Figure 2d) was observed. At 3dpi, both infected groups had higher ILTV genome loads in the lungs when compared to the controls (*p* < 0.05–0.001). At 7 dpi, only the AB-S45 ILTV-infected group had higher ILTV genome loads in the lungs than the controls (*p* < 0.05).

### 3.3. Host Responses

#### 3.3.1. Recruitment of T Cells (CD4+ T Cells and CD8+ T Cells) and Macrophages

The recruitment of CD4+ and CD8+ T cells and macrophages in the trachea and lungs in infected and control chickens was quantified using the immunofluorescent technique. The representative images of the CD4+ and CD8+ T cells and macrophages in the trachea and lungs are shown in Figure 3 and Figure 4, respectively.

The recruitment of CD4+ T cells in the trachea of the wildtype (AB-S63) and CEO revertant ILTV (AB-S45)-infected chickens was higher than the uninfected controls at 3 and 7 dpi (*p* > 0.05–0.01) (Figure 3a). However, there was no significant differences in the recruitment of CD4+ T cells in the wildtype (AB-S63) and CEO revertant ILTV (AB-S45)-infected groups at 3 and 7 dpi (*p* > 0.05). Similarly, there was no significant difference in the recruitment of CD8+ T cells in the trachea of the wildtype ILTV (AB-S63) and the CEO revertant ILTV (AB-S45)-infected groups (*p* > 0.05). At 3 dpi, the CD8+ T cells recruitment was higher in the CEO revertant ILTV (AB-S45)-infected group when compared to the uninfected controls (*p* < 0.01). At 7 dpi, both infected groups showed higher recruitment of the CD8+ T cells when compared to the control group (*p* > 0.05) (*p* > 0.05–0.01) (Figure 3b). A significant recruitment of the macrophages was observed in the infected trachea tissues relative to the uninfected controls (*p* > 0.05–0.01) (Figure 3c).

At 3 and 7 dpi, the recruitment of CD4+ T cells in the lungs was higher in both infected groups compared to the uninfected controls (*p* < 0.05–0.01) (Figure 4a). Although no group differences in the CD8+ T cell recruitment in the lungs were observed at 3 dpi, the recruitment of the CD8+ T cells was higher in both ILTV-infected groups compared to the controls (*p* < 0.05–0.01) at 7 dpi (Figure 4b). The recruitment of the macrophages was significantly higher in the CEO vaccine revertant (AB-S45) ILTV-infected group when compared to the control group at 3 dpi (*p* < 0.01). However, the recruitment of the macrophages was increased in wildtype (AB-S63) ILTV-infected group when compared to the control group at 7 dpi (*p* < 0.01) (Figure 4c).

#### 3.3.2. IFN-γ, IL-1β, and iNOS mRNA Expressions in the Trachea and Lungs

The mRNA expressions of IFN-γ, IL-1β, and iNOS were quantified in relation to the β actin housekeeping gene. The mRNA expression of cytokines and iNOS in infected tissues was calculated based on the mRNA expression of the control tissues at each time point. A significant difference in the mRNA expressions of IFN-γ and IL-1β (Figure 5a,b) was not observed in the trachea between 3 dpi and 7 dpi and between groups (*p* > 0.05). The iNOS mRNA expression (Figure 5c) was lower (*p* = 0.0043) at 7 dpi in comparison to that observed in 3 dpi in chickens of both infected groups. In the lungs, no difference in the IFN-γ mRNA expression (Figure 5d) was observed between 3 and 7 dpi in both AB-S45 and AB-S63 ILTV infected groups or between groups (*p* > 0.05). At 3dpi, the IL-1β mRNA expression (Figure 5e) was higher in the lungs of the AB-S63 ILTV-infected chickens compared to that observed in the lungs of the AB-S45 ILTV-infected chickens (*p* < 0.05). The iNOS mRNA expression (Figure 5f) remained unchanged in the lungs of the AB-S45 ILTV-infected chickens between 3 and 7 dpi (*p* > 0.05). At 7dpi, the iNOS mRNA expression (Figure 5f) was higher in the lungs of the AB-S63 ILTV-infected chickens when compared to that observed in the lungs of the AB-S45 ILTV-infected chickens (*p* < 0.05).

#### 3.3.3. Histopathological Findings in the Trachea

The representative images of tracheal lesions seen in each group are shown in Figure 6. In general, the histological lesions are severe at 3 dpi compared to 7 dpi. Substantial inflammation was observed in the tracheal mucosa of wildtype (AB-S63) and CEO vaccine revertant (AB-S45) ILTV-infected chickens when compared to the controls. The loss of cilia was also noted in both controls and infected groups but often with greater severity in the ILTV-infected chickens. The tracheal lesion scores showed no differences among groups and between time points (*p* ˃ 0.05).

##### Quantitative Histopathological Scoring of Tracheal Lesions

The histopathological lesion score of the trachea tissues between groups at each time point were compared using the Kruskal–Wallis test with Dunn’s Multiple Comparison Test, to identify groups (Table 1). Two-way ANOVA with Tukey’s multiple comparison were employed for the tracheal mucosal thickness analysis (Table 2). No significant difference in the histological lesion scores of the tracheal mucosal thickness between groups or time points was noted.

## 4. Discussion

Worldwide, CEO vaccine revertant strains have been frequently recovered from severe ILT outbreaks due to the widespread use of CEO vaccines in the ILT control [6,25,27,43,44,45,46]. Although most of the Canadian ILTV field strains associated with ILT outbreaks are genetically related to the CEO vaccine revertant strains [6,25,33], information is scarce on the pathogenesis and host responses relevant to the CEO vaccine revertant ILTV [32,34,47]. In the current study, we investigated the pathogenesis of, and the host responses to, Canadian-origin wildtype (AB-S63) and CEO vaccine revertant (AB-S45) ILTV isolates, and our findings are several. First, we observed that the ILTV isolates, the wildtype (AB-S63) and CEO vaccine revertant (AB-S45), are highly pathogenic, as demonstrated by clinical manifestations. Second, we found that the CEO vaccine revertant ILTV isolate (AB-S45) was replicated in primary replication sites better than wildtype ILTV (AB-S63), as evidenced by higher ILTV genome loads in OP swabs and the trachea. Third, we did not see that this difference in ILTV infection in the primary replication site between wildtype and the CEO vaccine revertant ILTV reflected in immune cell recruitment. Finally, we show that IL-1β and iNOS mRNA expressions in the lungs of wildtype ILTV-infected chickens were higher than that observed in the CEO vaccine revertant ILTV.

The infecting ILTV strain has an impact on the severity of the disease [9,14]. In the current study, characteristic clinical and pathological changes in the trachea induced by wildtype (AB-S63) and CEO vaccine revertant (AB-S45) ILTV isolates demonstrated severe respiratory infection. However, at the peak of the clinical signs (3 and 4 dpi), the wildtype (AB-S63) isolate obtained a significantly higher clinical scores than the CEO vaccine revertant (AB-S45) ILTV isolate. In contrast, based on the severity of the clinical signs, the pathology and the mortality rate of the previous studies have demonstrated that the ILTV strains related to the CEO vaccines are more pathogenic than the wildtype ILTV strains [9,34,35,48].

It has been shown that there is a positive correlation between the scores for microscopic tracheal lesions and viral genome loads in the trachea of the ILTV-infected chickens, which indicates viral lytic replication in this tissue [12]. However, a significantly higher genome loads of CEO vaccine revertant ILTV isolate AB-S45 was found in OP swabs and trachea than wildtype ILTV AB-S63 at 3 and 7 dpi. In agreement with a previous study [34], this result suggests better replication of the CEO vaccine revertant ILTV isolate AB-S45 in the primary replication sites. The difference between isolates in replication could be due to the higher affinity of the vaccine strains, particularly CEO vaccine related strains, for the tracheal tissue than the field strains [12]. Moreover, it is believed that the presence and persistence of the CEO vaccine revertant ILTV with a higher concentration in swabs and tissue could lead to greater transmission potential when compared to the wild type ILTV [34].

It has been reported that cell-mediated immunity plays a major role in the control of ILTV infection than antibody-mediated immune responses [29,30]. In the current study, the recruitment of T cells and macrophages were observed in the trachea and lungs of chickens infected with wildtype and CEO vaccine revertant ILTV isolates from 3 dpi until 7 dpi. As AB-S63 and AB-S45 ILTV isolates were propagated in CAM and CELICs, respectively, the component of chicken eggs and medium would affect the immune response; however, in the current experiment the titer of the stock AB-45 and AB-63 ILTV were high, and a very small amount of stock virus was diluted in PBS for infection; hence, the effect of embryo and media components in the immune response could be negligible. Although there are no past studies that assessed and compared the recruitment of T cells and macrophages in the tissues following infection with wildtype and CEO vaccine revertant ILTV, our observation is in agreement with a previous study [49], where infiltrations of CD4+ and CD8+ T cells, and macrophages were observed in the trachea of ILTV strain 1874C5 infected chickens. Similarly, another study [50] reported that at the late stage of ILTV infection, the transcription of genes associated with T cell populations (CD4+ and CD8+) was higher in the trachea tissues. This result could explain the decrease in the ILTV replication at 7 dpi, as indicated by the viral genome loads in the swabs and the tissue of the two ILTV infected groups. It has been also observed that the increased infiltration of the CD4+ T cells to the trachea of the chickens after ILTV infection could be as a result of the need to activate or regulate other immune cells through cytokine production, such as IFN-γ, and control the severe infection [49,51]. Meanwhile, macrophages are acknowledged as multipurpose cells that function as both effectors and modulators of innate immunity. They play a pivotal role in phagocytosis, in antigen presentation, and in regulating the activity of the adaptive immune response at the tissue level [52,53]. Similar to ILTV infection, macrophage recruitment to the trachea has also been observed following infection with other avian respiratory pathogens, such as infectious bronchitis virus (IBV) [54] and H4N6 low pathogenic avian influenza virus (LPAIV) [55].

In the current study, although no significant differences were detected between ILTV infected groups in the cytokine mRNA expressions in the trachea, we observed that IL-1β and iNOS mRNA expressions in the lungs of wildtype ILTV-infected chickens were higher than those observed in the CEO vaccine revertant ILTV. We could not find the literature regarding the cytokine transcription profiling of CEO vaccine revertant strains. However, our observations are in line with a previous study [56], where the ILTV infection of chicken embryo lung cells showed the transcription upregulation of IL-1β. It is believed that lung macrophages could be the main source of IL-1β and iNOS [39,57]. Since a significant recruitment of the macrophages in the lungs following ILTV infection with wildtype (AB-S63) ILTV isolate was observed in this study, it is possible that the lung macrophage played an important role in the upregulation of IL-1β and iNOS, which resulted in decreased viral replication [40,58,59]. Similarly, IL-1β and iNOS upregulations were also recorded following infection with other avian respiratory pathogens, such as IBV [39,57] and LPAIV [60,61].

The statistical significance of the clinical score, weight gain, and ILTV genome loads in swabs between AB-45 and AB-63 ILTV of the current study were different from our previous study [34]. It has been shown that the variations in observations could occur in animal studies, for various reasons [62]. Further, there are differences in the sampling time points, the number of groups with different viral strains, and the number of birds per group in between the current study and the previous experiment. Furthermore, the difference in sample size and the number of groups to be compared would affect the significance of the group differences.

## 5. Conclusions

Taken together, this study provides a better understanding of the host responses mounted against circulating Canadian-origin ILTV strains. Our data also indicates that the CEO vaccine revertant ILTV isolate, AB-S45, is as pathogenic as the wildtype ILTV isolate, AB-S63. However, further investigations are essential to confirm our observations and to establish the efficacy of the currently available ILT vaccines in the control of the CEO vaccine revertant isolates in the field.

## Figures and Tables

**Figure 1 vaccines-10-00782-f001:**
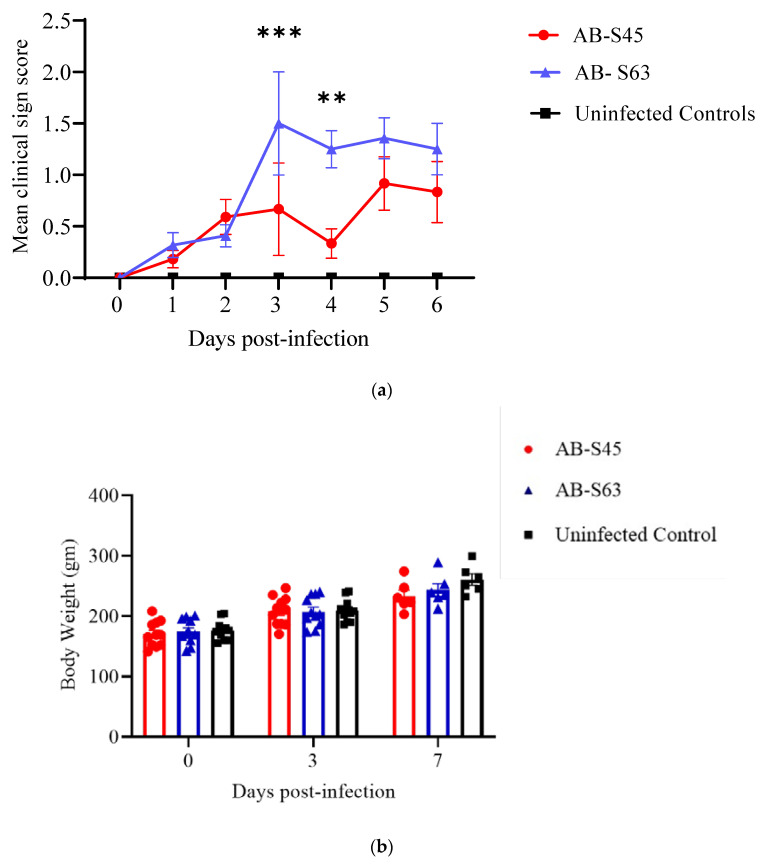
The clinical scores and body weights of chickens following infection with wildtype (AB-S63) and chicken-embryo-origin (CEO) vaccine revertant (AB-S45) ILTV isolates. The means of the clinical scores observed from 0–6 dpi and the body weights recorded on days 0, 3, and 7 following ILTV infection are plotted in (**a**,**b**), respectively. The error bars represent the standard error of means (SEM). The clinical signs were scored as described in the material and methods and statistically analyzed by employing the Kruskal–Wallis and Dunn’s multiple comparison tests. The differences in body weights between the groups were analyzed with the two-way ANOVA and Tukey’s multiple comparison tests. ** *p* < 0.01, *** *p* < 0.001.

**Figure 2 vaccines-10-00782-f002:**
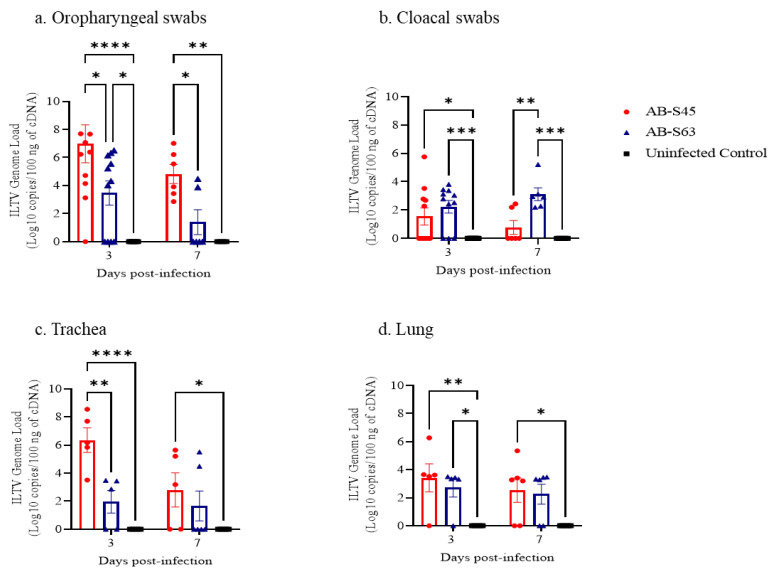
The ILTV genome loads at 3 and 7 dpi following infection with wildtype (AB-S63) and CEO vaccine revertant (AB-S45) ILTV isolates. ILTV genome loads in (**a**) oropharyngeal swabs; (**b**) cloacal swabs; (**c**) lung; and (**d**) trachea are illustrated. The Kruskal–Wallis test, followed by Dunn’s multiple comparison test, was employed to identify group differences in ILTV genome load. The error bars indicate the SEM. * *p* < 0.05, ** *p* < 0.01, and *** *p* < 0.001, **** *p* < 0.0001.

**Figure 3 vaccines-10-00782-f003:**
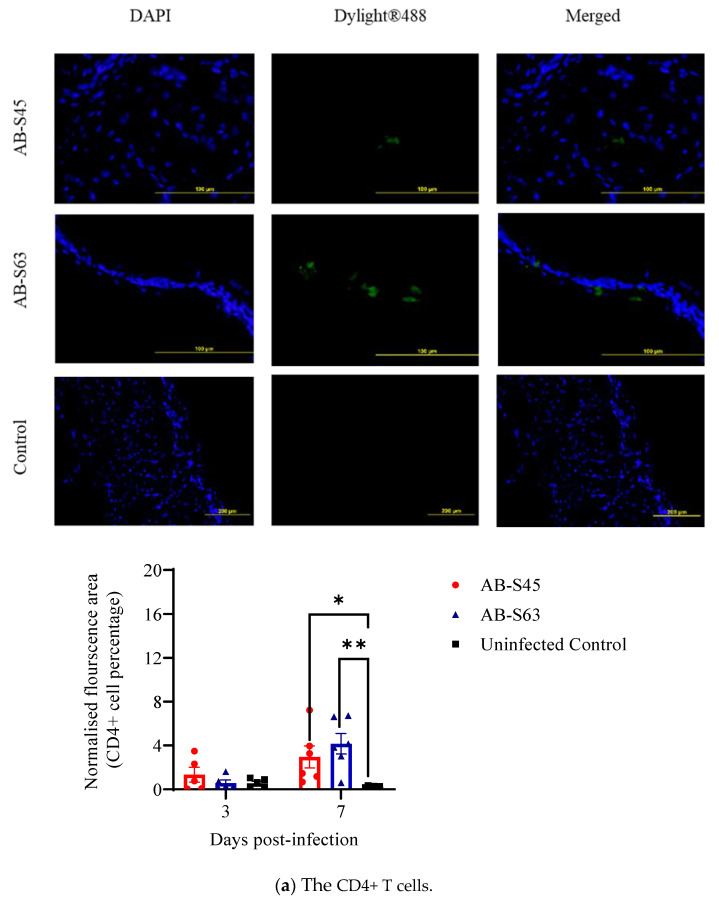
The immune cell recruitment in infected and control trachea following infection with wildtype (AB-S63) and CEO vaccine revertant (AB-S45) ILTV isolates and mock infection with PBS. The representative images show (**a**) CD4+ T cells, (**b**) CD8+ T cells, and (**c**) macrophages present in the trachea of the infected and uninfected chickens along with quantitative data. The Mann–Whitney U test was employed to compare infected and uninfected groups at 3 and 7 dpi. The error bars indicate the SEM. Significance: * *p* < 0.05, ** *p* < 0.01.

**Figure 4 vaccines-10-00782-f004:**
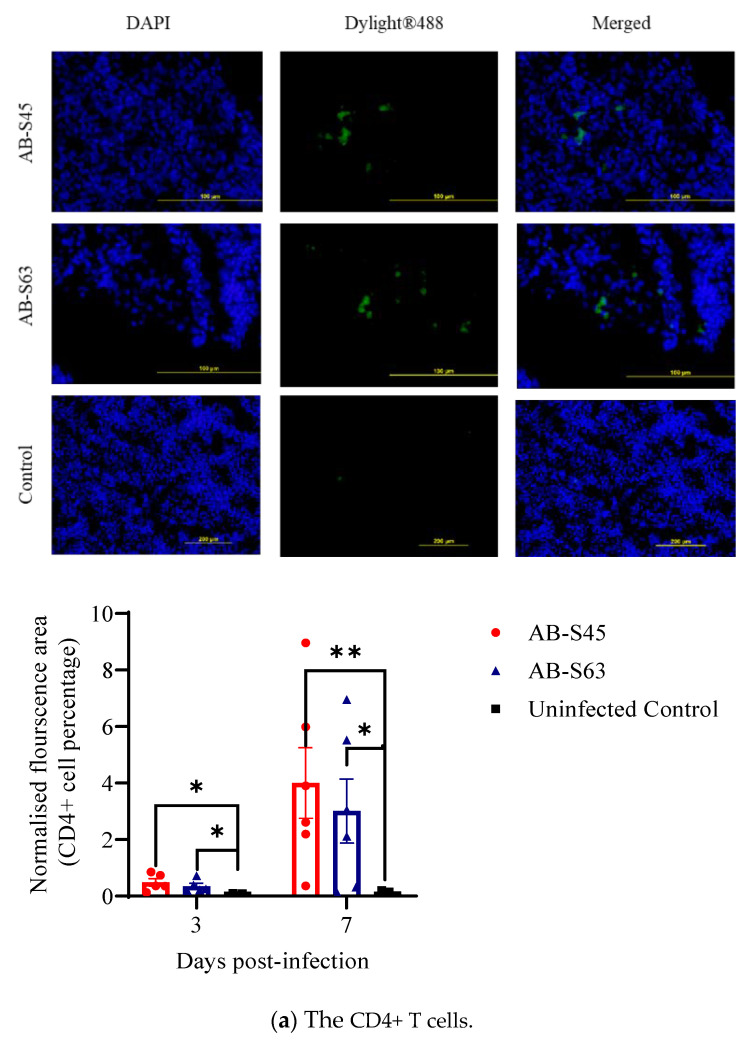
The immune cell recruitment in infected and control lungs following infection with wildtype (AB-S63) and CEO vaccine revertant (AB-S45) ILTV isolates and mock infection with PBS. The representative images show (**a**) CD4+ T cells, (**b**) CD8+ T cells, and (**c**) macrophages present in the lung tissue of the infected and uninfected chickens, along with quantitative data. The Mann–Whitney U test was employed to compare the infected and uninfected groups at 3 and 7 dpi. The error bars indicate the SEM. Significance: * *p* < 0.05, ** *p* < 0.01.

**Figure 5 vaccines-10-00782-f005:**
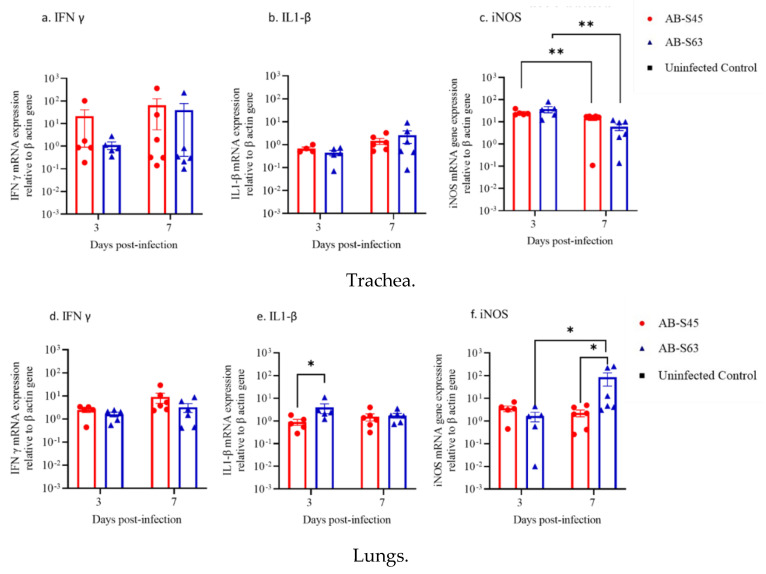
The immune gene mRNA expressions after infection with the wildtype (AB-S63) and CEO vaccine revertant (AB-S45) ILTV isolates. IFN-γ, IL-1β, and iNOS mRNA expressions in the trachea at 3 and 7 dpi is shown in (**a**–**c**), respectively, and in the lungs at 3 and 7 dpi are illustrated in (**d**–**f**), respectively. The Mann–Whitney U test was used to analyze the differences between time points and between groups. The error bars represent the SEM. Significance: * *p* < 0.05, ** *p* < 0.01.

**Figure 6 vaccines-10-00782-f006:**
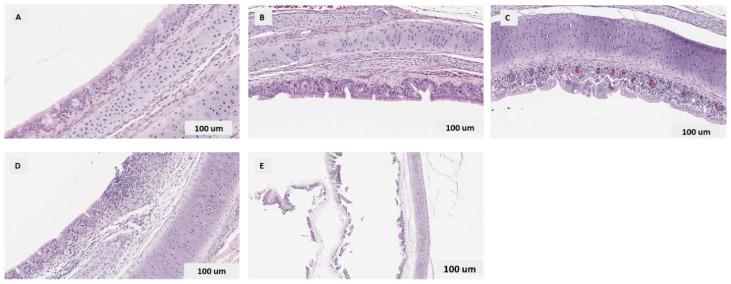
The representative images of the histology of the trachea from ILTV-infected chicken and controls. (**A**) A normal trachea with pseudostratified ciliated columnar epithelium and normal distribution of mucous glands, score 0; (**B**) minimal changes with normal epithelium plus mild to moderate infiltration of lymphocytes, rare heterophils, normal mucous glands, without syncytia or intranuclear inclusions, score 1; (**C**) mild changes with thickened mucosa with mild to moderate cell infiltration and/or normal epithelium, except for foci of syncytia, intranuclear inclusions, and hyperemia, occasionally with cell cuffs, score 2; (**D**) moderate changes with thickened mucosa with moderate to marked cell infiltration, numerous syncytia, intranuclear inclusions, patches of separating or sloughing epithelium from the lamina propria, the mucosal surface well covered by normal or affected epithelium, mucous glands reduced, marked hyperemia, and mononuclear cell infiltration around blood vessels, score 3; (**E**) severe changes with thickened mucosa, edema, proteinaceous fluid, cellular exudate, or adherent fibrinohemorrhagic to cellular pseudomembrane on the surface, normal epithelium absent, the mucosal surface covered by a thin layer of basal cells, and syncytia with intranuclear inclusions sometimes present, score 4.

**Table 1 vaccines-10-00782-t001:** The lesion scoring of histopathological lesions in the tracheas of chickens at 3 and 7 days after exposure to wildtype (AB-S63) and CEO vaccine revertant (AB-S45) ILTV isolates.

Bird #	3 Days Post-Infection			7 Days Post-Infection		
	Control	AB-S45 ILTV	AB-63 ILTV	Control	AB-S45 ILTV	AB-S63 ILTV
1	1	3	4	0	4	1
2	0	1	2	1	3	1
3	0	2	1	1	1	1
4	1	3	2	1	1	5
5	1	1	1	0	1	0
6	1	-	-	1	n/a *	3
Mean ± SEM **	0.6 ± 0.25	2 ± 0.45	2 ± 0.55	0.7 ± 0.2	2 ± 0.7	1.8 ± 0.8

* n/a: not enough sample to assess. **: the standard error of means.

**Table 2 vaccines-10-00782-t002:** The mucosal thickness in the tracheas of chickens at 3 and 7 days after exposure to the AB-S45 and AB-S63 ILTV isolates.

Bird #	3 Days Post-Infection			7 Days Post-Infection		
	Control	AB-S45 ILTV	AB-63 ILTV	Control	AB-S45 ILTV	AB-S63 ILTV
1	56.85	60.51	21.74	29.60	44.20	33.13
2	60.91	28.13	34.48	23.42	66.86	29.34
3	42.71	27.46	35.35	30.85	37.34	25.84
4	58.25	36.20	21.38	34.21	41.25	24.04
5	25.97	67.06	19.63	36.52	39.68	52.97
6	26.63	-	-	20.76	n/a *	48.32
Mean± SEM **	45.2 ± 6.5	43.87 ± 8.3	26.51 ± 3.4	29.23 ± 2.5	45.87 ± 5.4	35.61 ± 4.9

* n/a: not enough sample to assess. **: the standard error of means.

## Data Availability

Not applicable.

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
