# Peer review of "Host Responses Following Infection with Canadian-Origin Wildtype and Vaccine Revertant Infectious Laryngotracheitis Virus"

_vaccines, 2022, doi:10.3390/vaccines10050782_

Round 1

Reviewer 1 Report

Major comments:

  1. Some results were inconsistent with the results in the previous study performed by the same group (Viruses. 2021 Mar 24;13(4):541.). Details are described below. The authors should describe and discuss why those differences were occurred.

  • The authors showed the clinical sign scores of chickens of uninfected controls and infected with AB-S63 and AB-S45 in Figure 1a. The authors described the clinical score of AB-S63 was significantly higher than AB-S45 in this manuscript, however, there were no significant difference between the clinical scores of AB-S63 and AB-S45 in Figure 1a of the previous study.
  • The authors showed the weight of chickens of uninfected controls and infected with AB-S63 and AB-S45 in Figure 1b. The authors described there were no significant difference between the weight of the uninfected control, AB-S63 and AB-45 in this manuscript. However, the weight gain of AB-S63 and AB-S45 were significantly decreased compared to the uninfected control on 7 days post-infection in Figure 2a of the previous study. Were there significant differences in weight gain between the uninfected control, AB-S63 and AB-S45 in this study?
  • The authors showed the ILTV genome loads in oropharyngeal swabs of chickens of uninfected controls and infected with AB-S63 and AB-S45 in Figure 2a. The mean of ILTV genome load of AB-S45 infected chickens on day 3 in this manuscript were similar to that in Figure 4a of the previous study. The mean of ILTV genome load of AB-S63 infected chickens on day 3 in this manuscript seems 2-3 log lower than that in the previous study. The mean of ILTV genome load of AB-S45 infected chickens on day 7 in this manuscript seems >10 log lower than that in the previous study. The mean of ILTV genome load of AB-S63 infected chickens on day 7 in this manuscript seems 3 log lower than that in the previous study.
  • The authors showed the ILTV genome loads in cloacal swabs of chickens of uninfected controls and infected with AB-S63 and AB-S45 in Figure 2b. The mean of ILTV genome load of AB-S45 infected chickens on day 3 in this manuscript seems 4 log lower than that in Figure 5a of the previous study. The mean of ILTV genome load of AB-S63 infected chickens on day 3 in this manuscript seems 1-2 log higher than that in the previous study. The mean of ILTV genome load of AB-S45 infected chickens on day 7 in this manuscript seems 5 log lower than that in the previous study. The mean of ILTV genome load of AB-S63 infected chickens on day 7 in this manuscript seems 2 log lower than that in the previous study. The authors also stated that ILTV genome loads in AB-S63 infected group was significantly higher than that of AB-S45 infected group in this manuscript, however, there were no significant difference in ILTV genome loads between AB-S63 infected group and AB-S45 infected group.

  1. The authors inoculated PBS for uninfected controls. Were the propagated viruses diluted with PBS? As AB-S63 and AB-S45 were propagated in CAM and CELICs, respectively, the component of chicken eggs and medium would affect immune response. The authors should discuss about this point.

Minor comments:

  1. There were some typographical errors in the manuscript, for example, in line 121.

Author Response

Major comments:

  1. Some results were inconsistent with the results in the previous study performed by the same group (Viruses. 2021 Mar 24;13(4):541.). Details are described below. The authors should describe and discuss why those differences were occurred.

Response:

In agreement with the comment, we included information (last paragraph in the discussion) as,

 “The statistical significance of clinical score, weight gain and ILTV genome load in swabs between AB-45 and AB-63 ILTV were different from our previous study. It has been shown that the variations in observations could happen in animal studies due to various reasons [1]. Further, there are differences in sampling time points, the number of groups with different viral strains and the number of birds per group between the current study and previous experiment. Furthermore, the difference in sample size and the number of groups to be compared would affect the significance of differences.

  1. von Kortzfleisch, V.T.; Karp, N.A.; Palme, R.; Kaiser, S.; Sachser, N.; Richter, S.H. Improving reproducibility in animal research by splitting the study population

  • The authors showed the clinical sign scores of chickens of uninfected controls and infected with AB-S63 and AB-S45 in Figure 1a. The authors described the clinical score of AB-S63 was significantly higher than AB-S45 in this manuscript, however, there were no significant difference between the clinical scores of AB-S63 and AB-S45 in Figure 1a of the previous study.

Response:

The previous experiment had 3 infected groups as AB-20, AB-S63 and AB-S45 with 8 birds per group. In the current experiment, there were only AB-45 and AB-63 ILTv groups with 11 birds in each group, since subset of birds euthanized at 3 dpi, the clinical score data was obtained from 6 birds/group from 4 dpi onwards. The discrepancy between statistical significance of clinical scores between the experiments could be due to difference in number of groups and number of birds.

  • The authors showed the weight of chickens of uninfected controls and infected with AB-S63 and AB-S45 in Figure 1b. The authors described there were no significant difference between the weight of the uninfected control, AB-S63 and AB-45 in this manuscript. However, the weight gain of AB-S63 and AB-S45 were significantly decreased compared to the uninfected control on 7 days post-infection in Figure 2a of the previous study. Were there significant differences in weight gain between the uninfected control, AB-S63 and AB-S45 in this study?

Response:

We didn’t observe a significant difference in weight gain between AB-S63 and AB-S45 groups as well.

  • The authors showed the ILTV genome loads in oropharyngeal swabs of chickens of uninfected controls and infected with AB-S63 and AB-S45 in Figure 2a. The mean of ILTV genome load of AB-S45 infected chickens on day 3 in this manuscript were similar to that in Figure 4a of the previous study. The mean of ILTV genome load of AB-S63 infected chickens on day 3 in this manuscript seems 2-3 log lower than that in the previous study. The mean of ILTV genome load of AB-S45 infected chickens on day 7 in this manuscript seems >10 log lower than that in the previous study. The mean of ILTV genome load of AB-S63 infected chickens on day 7 in this manuscript seems 3 log lower than that in the previous study.

Response:

The previous experiment had 8 birds per group and the data was obtained from all 8 birds at each time point as there were no euthanasia in between. In the current experiment, we had 11 birds per group and 5 and 6 birds from each group were euthanized at 3 dpi and 7 dpi respectively. Since there can be individual variation in genome load, the number of birds in each group could be the reason for difference in mean genome load values.

  • The authors showed the ILTV genome loads in cloacal swabs of chickens of uninfected controls and infected with AB-S63 and AB-S45 in Figure 2b. The mean of ILTV genome load of AB-S45 infected chickens on day 3 in this manuscript seems 4 log lower than that in Figure 5a of the previous study. The mean of ILTV genome load of AB-S63 infected chickens on day 3 in this manuscript seems 1-2 log higher than that in the previous study. The mean of ILTV genome load of AB-S45 infected chickens on day 7 in this manuscript seems 5 log lower than that in the previous study. The mean of ILTV genome load of AB-S63 infected chickens on day 7 in this manuscript seems 2 log lower than that in the previous study. The authors also stated that ILTV genome loads in AB-S63 infected group was significantly higher than that of AB-S45 infected group in this manuscript, however, there were no significant difference in ILTV genome loads between AB-S63 infected group and AB-S45 infected group.

Response:

As mentioned before the number of birds in each group and individual variation in the genome load could be the reason for the difference in mean genome load of AB-45 and AB-63 between both experiments. In response to difference in statistical significance between genome load of both experiments, the previous experiment had 3 infected groups with one time point and the current experiment has 2 infected groups with two time points. The sample size and the number of comparisons would affect the results of statistical analysis. This could be a reason for significant difference in between groups.

  1. The authors inoculated PBS for uninfected controls. Were the propagated viruses diluted with PBS? As AB-S63 and AB-S45 were propagated in CAM and CELICs, respectively, the component of chicken eggs and medium would affect immune response. The authors should discuss about this point.

 Response

The virus was diluted in PBS for infection. The stock virus concentration was very high, for example for AB 63 the stock titer was 107 TCID50. Therefor 0.3 µl from the stock virus was diluted in 199.7 µl of PBS for infection per bird. Hence, the effect of chicken and media components in immune response could be minimum.

In agreement with the comment a sentence was included in the discussion section (4th paragraph) as

‘As AB-S63 and AB-S45 were propagated in CAM and CELICs, respectively, the component of chicken eggs and medium would affect the immune response, however in the current experiment the titer of the stock AB-45 and AB-63 viruses were high and a very small amount of stock virus was diluted in PBS for infection, hence, the effect of embryo and media components in immune response could be negligible.’

Minor comments:

  1. There were some typographical errors in the manuscript, for example, in line 121.

Response

In agreement, we addressed the errors.

Reviewer 2 Report

This is an interesting manuscript. The fact that CEO is responsible for ILT outbreaks world wide is known for several years. The information presented in this manuscript is valuable and of scientific interest. 

ILTV infection causes significant economic losses for the poultry sector around the world due to high mortalities and production losses. Presently, vaccination is done with live attenuated vaccines and viral vector recombinant vaccines. The live attenuated vaccines are still acceptable choices for managing the disease as they confer the highest level of protection against ILTV infection by decreasing both clinical signs and viral replication more efficiently than the recombinant vaccines. Despite their efficiency, attenuated vaccines, especially chicken embryo origin (CEO) vaccines, have adverse properties of regaining virulence following bird to bird passages leading to development of virulent ILTV vaccinal strains in the field. Recombination between different ILTVs, including vaccine strains, leading to the emergence of highly virulent field strains.

The aim of this study was to compare the pathogenesis of, and host responses to Canadian origin CEO revertant and wildtype ILTV infection in 
chickens. The authors investigated the pathogenesis of, and the host responses to Canadian origin wildtype (AB-S63) and CEO vaccine revertant (AB-S45) ILTV isolate, and their findings are : First, they recorded that the wildtype (AB-S63) and CEO vaccine revertant (AB-S45) ILTV isolates are highly pathogenic as demonstrated by clinical manifestations. Second, they found that CEO vaccine revertant ILTV isolate, AB-S45 replicated in primary replication sites better than wildtype ILTV, AB-S63 as evidenced by higher ILTV genome loads in oropharyngeal swabs and trachea. Third, they did not see that this difference in ILTV infection in primary replication site between wildtype and CEO vaccine revertant ILTV reflected in immune cell recruitment. Finally, they show that IL-1β and iNOS mRNA expressions in lungs of wildtype ILTV infected chickens were higher than that observed in CEO vaccine revertant ILTV.

Author Response

This is an interesting manuscript. The fact that CEO is responsible for ILT outbreaks world wide is known for several years. The information presented in this manuscript is valuable and of scientific interest. 

ILTV infection causes significant economic losses for the poultry sector around the world due to high mortalities and production losses. Presently, vaccination is done with live attenuated vaccines and viral vector recombinant vaccines. The live attenuated vaccines are still acceptable choices for managing the disease as they confer the highest level of protection against ILTV infection by decreasing both clinical signs and viral replication more efficiently than the recombinant vaccines. Despite their efficiency, attenuated vaccines, especially chicken embryo origin (CEO) vaccines, have adverse properties of regaining virulence following bird to bird passages leading to development of virulent ILTV vaccinal strains in the field. Recombination between different ILTVs, including vaccine strains, leading to the emergence of highly virulent field strains.

The aim of this study was to compare the pathogenesis of, and host responses to Canadian origin CEO revertant and wildtype ILTV infection in chickens. The authors investigated the pathogenesis of, and the host responses to Canadian origin wildtype (AB-S63) and CEO vaccine revertant (AB-S45) ILTV isolate, and their findings are: First, they recorded that the wildtype (AB-S63) and CEO vaccine revertant (AB-S45) ILTV isolates are highly pathogenic as demonstrated by clinical manifestations. Second, they found that CEO vaccine revertant ILTV isolate, AB-S45 replicated in primary replication sites better than wildtype ILTV, AB-S63 as evidenced by higher ILTV genome loads in oropharyngeal swabs and trachea. Third, they did not see that this difference in ILTV infection in primary replication site between wildtype and CEO vaccine revertant ILTV reflected in immune cell recruitment. Finally, they show that IL-1β and iNOS mRNA expressions in lungs of wildtype ILTV infected chickens were higher than that observed in CEO vaccine revertant ILTV.

Response: Thank you for the comments.

Round 2

Reviewer 1 Report

It is true that the variations in observations could happen in animal studies and the difference in sample size and the number of groups to be compared would affect the significance of differences. However, some discrepancy cannot be explained by the difference of sample  size, the number of groups to be compared and so on.

As I mentioned before, the mean of ILTV genome load of AB-S63 infected chickens on day 3 in this manuscript seems 2-3 log lower than that in the previous study. The mean of ILTV genome load of AB-S45 infected chickens on day 7 in this manuscript seems >10 log lower than that in the previous study. The difference of viral genome loads between the previous and this study was too big to be explained by the reasons presented by the authors. The author said the sample size was N-8 and 5-6 in the previous and this study, respectively, which caused difference in mean genome load values. If so, the birds euthanized on day 3 and 7, which used for immune response analysis and pathogenesis analysis, might had very low or high viral load compared to the birds used to detect the viral load. As the differences in viral load would affect the infiltration of immune cells and pathogenesis, the result and conclusion would be changed.

If the authors do not show convincing arguments, it might be difficult to be accepted.

Author Response

Reviewer comments.

It is true that the variations in observations could happen in animal studies and the difference in sample size and the number of groups to be compared would affect the significance of differences. However, some discrepancy cannot be explained by the difference of sample size, the number of groups to be compared and so on.

As I mentioned before, the mean of ILTV genome load of AB-S63 infected chickens on day 3 in this manuscript seems 2-3 log lower than that in the previous study. The mean of ILTV genome load of AB-S45 infected chickens on day 7 in this manuscript seems >10 log lower than that in the previous study. The difference of viral genome loads between the previous and this study was too big to be explained by the reasons presented by the authors. The author said the sample size was N-8 and 5-6 in the previous and this study, respectively, which caused difference in mean genome load values. If so, the birds euthanized on day 3 and 7, which used for immune response analysis and pathogenesis analysis, might had very low or high viral load compared to the birds used to detect the viral load. As the differences in viral load would affect the infiltration of immune cells and pathogenesis, the result and conclusion would be changed.

If the authors do not show convincing arguments, it might be difficult to be accepted.

Response: We agree with comments of the reviewer and made changes to the discussion and conclusions made. The changes are highlighted in the abstract, discussion, and conclusions sections.